# Log-FGAER: Logic-Guided Fine-Grained Address Entity Recognition from Multi-Turn Spoken Dialogue

**Xue Han**[1*]**, Yitong Wang**[1*]**, Qian Hu**[1]**, Pengwei Hu**[2]**, Chao Deng**[1]**, Junlan Feng**[1] ✉

[1]JIUTIAN Team, China Mobile Research Institute, Beijing, China
[2]The Xinjiang Technical Institute of Physics and Chemistry, Urumqi, China

{hanxueai, wangyitongyjy, huqianai, dengchao, fengjunlan}@chinamobile.com
hupengwei@hotmail.com

## Abstract

Fine-grained address entity recognition (FGAER) from multi-turn spoken dialogues is particularly challenging. The major reason lies in that a full address is often formed through a conversation process. Different parts of an address are distributed through multiple turns of a dialogue with spoken noises. It is nontrivial to extract by turn and combine them. This challenge has not been well emphasized by main-stream entity extraction algorithms. To address this issue, we propose in this paper a logic-guided fine-grained address recognition method (Log-FGAER), where we formulate the address hierarchy relationship as the logic rule and softly apply it in a probabilistic manner to improve the accuracy of FGAER. In addition, we provide an ontology-based data augmentation methodology that employs ChatGPT to augment a spoken dialogue dataset with labeled address entities. Experiments are conducted using datasets generated by the proposed data augmentation technique and derived from real-world scenarios. The results of the experiment demonstrate the efficacy of our proposal.

## 1 Introduction

Fine-grained address entity recognition from calls is an important task in many applications (Wu and Juang, 2022; Yang et al., 2022), such as obtaining delivery addresses from E-commerce assistant or post-sale service (Eligüzel et al., 2020). To be specific, the scenario considered in this paper is the extraction of fine-grained address entities distributed through the multi-turn spoken dialogue contexts. Figure 1 depicts an example of such a dialogue between a customer and an after-service call center in which the fine-grained address entities are recognized and then typically combined to form a full standard address for further delivery.

Existing name entity recognition (NER) methods usually combine the pre-trained language models (e.g. BERT) with supervised models such as BiLSTM/LSTM-CRF (Xu et al., 2019; Zhang et al., 2020; Mai et al., 2018) or with some expansive external knowledge base (Dogan et al., 2019). These methods, which are trained and evaluated on text from well-written sources, do not perform well in a spoken dialogue context, because the entities in the spoken scenario are distributed across a multi-turn context with a lot of noise. Other studies explore the NER problem in a spoken or dialogue context (Muralidharan et al., 2021; Chen et al., 2022; Luoma and Pyysalo, 2020; Hanh et al., 2021), but their labels are not applicable to the address domain. Because of the diversity and complexity of multi-turn spoken dialogue scenarios, annotating for the FGAER task is laborious and time-consuming (Shan et al., 2020).

To address these issues, we propose a logic-guided fine-grained address entity recognition method (Log-FGAER). The Log-FGAER is based on the fact that address entities follow spatially constrained relationships (Zandbergen, 2008). As a result, the multi-turn spoken dialogue must maintain a discourse structure to get the full standard address for the duration of address acquiring in accordance with the spatial constraint relationship between the geographical address entities. Figure 1(a) shows an example of a 4-level spatial dependency tree structure, in which each node is a fine-grained address entity, each level represents an entity type (e.g. "city" and "district"), and each edge is a dependency relation between the entities. A full standard address is made up of nodes in the path from the root to the leaf node. We can improve entity recognition inference by presenting such an interdependent hierarchy relationship as the logic rule and then using probabilistic soft logic (PSL) to regulate the loss objective.

We further introduce an ontology-based data aug-

---

* These authors contributed to the work equally.
✉ The corresponding author.

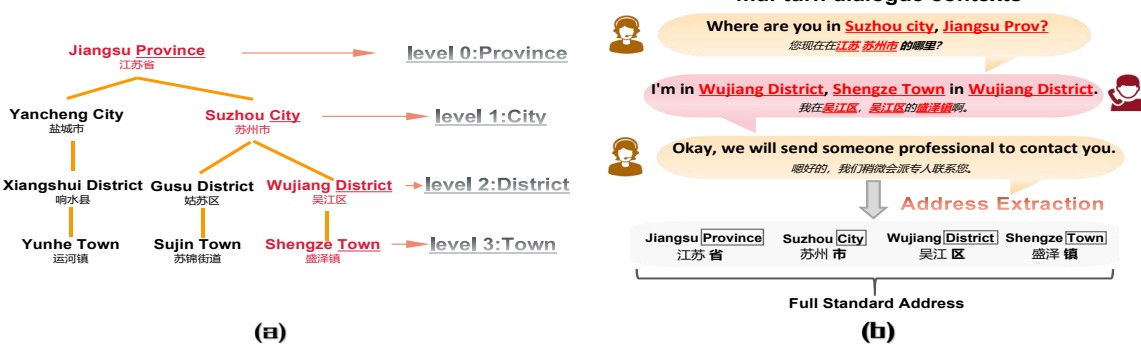

Figure 1: An example of location extraction task from a multi-turn dialogue. The path in red in (a) reflects the hierarchy relationship of the address entities extracted from (b).

mentation paradigm to build a labeled multi-turn dialogue dataset for the FGAER task, which could inspire future research. The paradigm first extracts utterance and response (UR) templates from publicly available dialogue datasets and then leverages the largely available full standard address to get labeled fine-grained address entities. The address entities in the UR templates could be altered using the technique we propose to expand current dialogues. To imitate real-world conversation circumstances, a noise injection approach and a ChatGPT augmentation method are also proposed.

We conduct experiments on both the created dataset and a small set of human-labeled datasets derived from real-world customer care conversations. Experimental results show that Log-FGAER improves baselines in both cases.

The contributions of this paper are summarized as follows:

- We make use of probabilistic soft logic to improve the FGAER tasks after utilizing logic rules to indicate the interdependent hierarchy relationship between fine-grained address entities, which reflects the discourse structure of an address-acquiring dialogue.

- We present an ontology-based and ChatGPT assisted data augmentation paradigm for generating a multi-turn spoken dialogue with labeled fine-grained address entities requiring little human effort, thereby addressing the issue of scarce labeled datasets and high labeling costs.

- Logic-guided method is a plug-and-play module compatible with all existing FGAER techniques. Experiments indicate that our Log-FGAER with PSL enhancement can obtain

competitive or even superior performance.

## 2 Related Works

### 2.1 Fine-grained Named Entity Recognition

Fine-grained named entity recognition aims to resolve the restriction of coarse-grained NER that produces labels from only a small set of entity classes. To solve the lack of datasets where entity boundaries are properly annotated while covering a large spectrum of entity types, (Dogan et al., 2019) proposes to combine learning models (ELMo) with an expansive knowledge base (Wikidata). (Mai et al., 2018) improves the neural network-based Japanese FG-NER performance by removing the CNN layer and utilizing dictionary and type embeddings. The most similar work with us is DLocRL (Xu et al., 2019), which trains a BiLSTM-CRF based POI recognizer for fine-grained location recognition and linking in tweets. This work only considers the POI locations, which is one of the entity types in our scenario.

### 2.2 Named Entity Recognition for dialogue

In dialogue systems, NER is a vital task for understanding user intent but suffers from the problems that NE labels are too coarse-grained or the entity types are not linked to a useful ontology. (Muralidharan et al., 2021) proposes a method of encoding the external knowledge of entities not directly using the toponym dictionary but by obtaining information from the knowledge graph. (Bowden et al., 2018) leverages external knowledge such as the Google Knowledge Graph API to solve the problems in the dialogue contexts. However, these methods have quite different entity types, which are not applicable for our scenario.

## 2.3 Probabilistic Soft Logic

In recent years, PSL rules have been applicable to various machine learning topics. In the knowledge graph construction area, (Chen et al., 2019) infers the confidence score of unseen relations by introducing PSL to represent the uncertain knowledge graph. In the temporal relation extraction topic, (Zhou et al., 2021) leverages PSL regularization to jointly train a relation classifier with sentence embeddings from a deep language model and perform the global temporal inference with a time graph. As for causal inference, (Du et al., 2021) acquires additional evidence information from a large-scale causal event graph as logical rules for causal reasoning. (Cai et al., 2022) proposes a neural-logic based soft logic enhanced event temporal reasoning model for acquiring unbiased temporal common sense knowledge. (Mohler et al., 2020) models the procedural state changes over time with PSL. Most of the works inject the logic knowledge to neural networks, by introducing logic-driven loss functions. These works encourage us to apply PSL rules to the FGAER task.

## 3 Method

### 3.1 Preliminary

**Problem Formulation.** The FGAER task is formulated as a sequence labeling problem (Xu et al., 2019). A multi-turn spoken dialogue is denoted as several turns of sequences $D = \{S_1, ..., S_i\}$, where $S_i$ represents one turn of question and answer in the dialogue. The FGAER's input $S_i$ is defined as $\{s_1, ..., s_j\}$, where $s_j$ denotes the $j_{th}$ token. The fine-grained address entities are spread across in $D$. The output is a tag sequence $Y_i = \{y_1, ...y_j\}$ for each input $S_i$, where $y_j \in \gamma$. $\gamma$ is the set of predefined tags following the BIOE tagging scheme. Each tag (e.g., B-city and I-district) in $\gamma$ is indicated by B-(begin), I-(inside), and E-(end) of a fine-grained address entity with its type in a type set. For a character that is not inside of any entity, its tag is O-(outside). The FGAER task aims at discovering a list of address entities $\{E^1, ..., E^n\}$. $E^n = \{s_{start}, ..., s_{end}\}$ is a substring of $S_i$ satisfying $start \leq end$.

**Probabilistic Soft Logic.** As mentioned before, the discourse structure of a multi-turn spoken dialogue usually reflects the spatial constraint relationship between the geographical address entities, which can be modeled with Probabilistic Soft Logic (PSL) (Kimmig et al., 2012). The PSL probabilistic

reasoning is incorporated as additional supervision signals to regulate the loss objective of FGAER task. We first introduce several concepts and notations in the PSL language following (Cai et al., 2022), and illustrate how the PSL rule is used to define templates for the hierarchical constraint dependency inference in FGAER task.

**Definition 1.** The **atom formula** $l$ is made up of a predicate $p$ together with its arguments. $l$ takes on continuous values in the unit interval $[0, 1]$.

We define three kinds of predicates for our scenario: HIGH, LOW, and BETWEEN. HIGH and LOW both take two arguments to work: an input sequence and the type of address entity. BETWEEN takes three arguments: an input sequence and two address entity types. For example, the atom $\text{HIGH}(S, city)$ represents that the highest level of address entity containing in the sequence $S$ is $city$. The atom $\text{BETWEEN}(S, city, town)$ means that the types of entities in $S$ are between $city$ and $town$ in the hierarchy structure.

**Definition 2.** **Complex formulae** are created by combining atom formulae with logical operations (e.g., conjunction ($\wedge$) and disjunction ($\vee$)).

**Definition 3.** A **logic rule** $r : f_1 \Rightarrow f_2$ is an implication formed by combining $f_1$ and $f_2$ with logical connectives. $f_1$ and $f_2$ can be atomic or complex formulae, respectively.

**Definition 4.** The soft truth values of the formula $f$ and the logic rule $r$ are denoted by the interpretations $\text{I}(f)$ and $\text{I}(r)$, which refer to the probability that $f$ and $r$ hold.

**Definition 5.** Łukasiewicz t-norms (Klir and Yuan, 1995) adopted to relax logic in PSL are defined as below. With the t-norms to relax logic, we can turn the logic rule ($r$) and simple or complex formulas ($f$) into differentiable functions, which can then be used as learning goals to control the loss objective: $I(f_1 \wedge f_2) = max(I(f_1) + I(f_2) - 1, 0)$ and $I(r : f_1 \Rightarrow f_2) = min(1, 1 - I(f_1) + I(f_2))$

With the above definitions, we can now define the logic rule that explain how the address entities in the current turn of a dialogue sequence relate to those in the previous and following turns.

### 3.2 Log-FGAER

**Definition 6.** Since multi-turn spoken dialogues maintain a discourse structure to get the full standard address during address acquisition, we can assume that the address entity types mentioned in

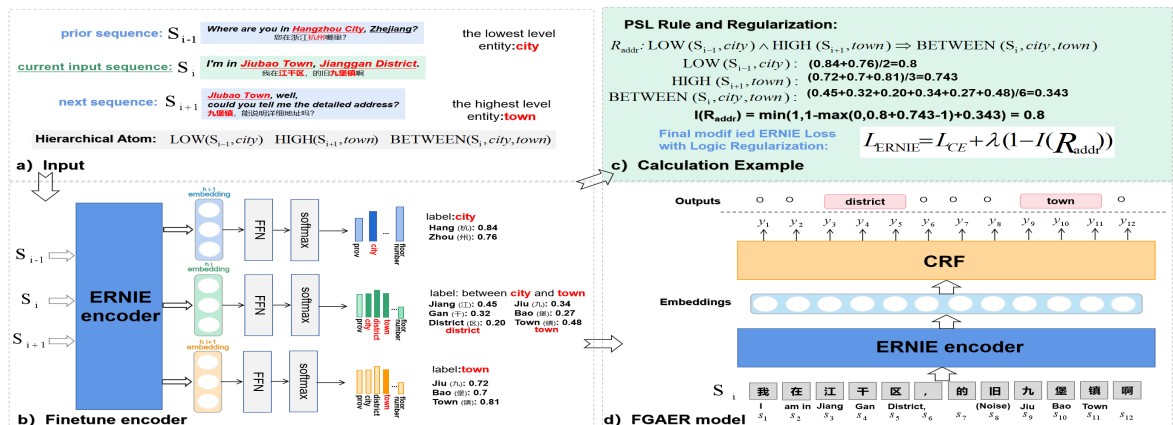

Figure 2: The main framework of Log-FGAER. a) The inputs of the encoder are: the current turn of the sequence and sequences before and after it. b)The ERNIE encoder is fine-tuned using PSL regularization to learn the spatial hierarchy constraints from the contexts for the current input sequence. c) An example of how to calculate the logic regularization. d) The embedding from the trained (PSL)ERNIE is then sent to the CRF layer for the FGAER task.

the current sequence $S_i$ are higher in the administrative division hierarchy than the highest level of address entity types in the next sequence $S_{i+1}$ but lower than the lowest level of types in the previous sequence $S_{i-1}$. Such a kind of relationship could be formulated as logic rule $\mathbf{R_{addr}} \overset{\Delta}{=} \mathbf{l_1} \wedge \mathbf{l_2} \Rightarrow \mathbf{f}$:

$$l_1 : LOW(S_{i-1}, a) \wedge l_2 : HIGH(S_{i+1}, b) \\ \Rightarrow f : BETWEEN(S_i, a, b) \quad (1)$$

In Eq 1, the atom formulae $l_1 : LOW(S_{i-1}, a)$ indicates that the lowest-level address type of $S_{i-1}$ is $a$. Similarly, $l_2 : HIGH(S_{i+1}, b)$ means that the highest-level address type of $S_{i+1}$ is $b$. The address entity types contained in $S_i$ are generally between $a$ and $b$ in the hierarchy structure. Because the hierarchical logic rule $\mathbf{R_{addr}}$ have the form $l_1 \wedge l_2 \Rightarrow f$, we can obtain a soft truth value by applying the Lukasiewicz t-norm to the interpretation of $\mathbf{R_{addr}}$.

Figure 2 illustrates the framework of Log-FGAER, which consists of two components: 1) a pre-trained language model (PLM) encoder that is fine-tuned with PSL regularization; and 2) an FGAER module that is composed of the fine-tuned PLM encoder and a CRF layer.

**1) Fine-tune the PLM encoder with PSL regularization.** In this component, the PSL rule from Definition 6 is used to calculate the logic regulation loss, which is then combined with the original PLM model's cross-entropy loss to fine-tune the PLM encoder. In this manner, the embeddings generated by the fine-tuned PLM encoder would incorporate the spatial constraint dependency information in the dialogue context.

More specifically, the PLM encoder accepts a sequence as input that may contain address entities and then outputs a token-level embedding of the sequence. The ERNIE (Sun et al., 2019) is proposed as the PLM backbone because it has been enhanced for named entity recognition by incorporating entity semantic information. Given $S_i$ as input, we could get the embedding $\mathbf{H_i} = ERNIE(\mathbf{S_i})$, where $H_i = \{h_1, ..., h_j\}$ and $h_j$ denotes the output of the pre-trained ERNIE that corresponds to the input token $s_j \in S_i$. The predicted probability distribution $\hat{y}_j$ for $s_j$ is then obtained by adding on top of the ERNIE a linear layer (Feed-Forward Network) with softmax, as shown below:

$$\hat{y}_j = P(y_j|s_j) = softmax(Wh_j + b) \quad (2)$$

In the above equation, $W$ and $b$ are trainable parameters. $y_j$ is the ground-truth label for $s_j$.

We compute the prediction loss $L_{ce}$ using the cross-entropy objective as below:

$$L_{ce} = -\frac{1}{|S_i|} \sum_{j=1}^{|S_i|} CrossEntropy(y_j, \hat{y}_j) \quad (3)$$

Following that, we will show how to compute the logic regulation loss $L_{logic}$ and then combine it with the $L_{ce}$ to ensure that the predicted distribution $\hat{y}_j$ is compatible with the hierarchical logic rule $\mathbf{R_{addr}}$, as shown in Definition 6.

To calculate $L_{logic}$, we should get $I(\mathbf{R_{addr}})$ first, which is the truth value of logic rule $\mathbf{R_{addr}}$. Figure 2(c) illustrates an $I(\mathbf{R_{addr}})$ calculation example. $I(LOW(S_i, city))$, also known as $I(l_1)$, is intended to be the predicted probabilities of the characters "Hangzhou city" to be the entity type of

"city". Similarly, I(BETWEEN($S_i, city, town$)), also known as I($f$), represents the predicted probabilities of "Jianggan District" and "Jiubao Town" in $S_i$, whose entity types are between "city" and "town". After getting the values of I($l_1$), I($l_2$) and I($f$), I($\mathbf{R_{addr}}$) which denotes the degree that the rule $\mathbf{R_{addr}}$ are satisfied by the predicted distribution, can be derived by incorporating the Lukasiewicz t-norm.

The principle of PSL is: the larger the truth values are, the better the logic rules are satisfied. Based on this principle, we formulate the distance of I($\mathbf{R_{addr}}$) to be true as a regularization term to penalize the distorted distribution that violates the logic rule $\mathbf{R_{addr}}$ (Cai et al., 2022). The logic regulation loss $L_{logic}$ is calculated as below:

$$L_{logic} = 1 - I(\mathbf{R_{addr}}) \tag{4}$$

We finalize the loss function by applying a weighted sum of Eq 3 and Eq 4:

$$L_{ERNIE} = L_{ce} + \lambda L_{logic} \tag{5}$$

where $\lambda$ is the weight for PSL regularization term. The objective is to minimize the $L_{ERNIE}$ and to finetune the ERNIE encoder parameters.

With the regularization of PSL rule on the outputs of the ERNIE encoder, our model can make embeddings capture more contextual information, which could further benefit the sequence labeling task described in the FGAER module.

**2) FGAER module** is made up of the ERNIE encoder that has been fine-tuned with PSL regulation and a CRF layer (Lafferty et al., 2001), denoting as (PSL)ERNIE-CRF. The loss function $L_{CRF}$ of FGAER is calculated by feeding $\mathbf{H}$ from Eq 2 into the CRF layer. The objective is to minimize $L_{CRF}$. The trained FGAER model could be used in the inference stage for the FGAER task.

## 4 Ontology Based Dialogue Data Augmentation Leveraging ChatGPT

### 4.1 Ontology-based dialogue enrichment

The goal of ontology-based dialogue enrichment is to address the issue of a shortage of dialogue datasets with labeled fine-grained address entities. As previously stated, labeling the fine-grained entities is costly and time-consuming. Fortunately, large-scale, unlabeled, full standard addresses (for example, Ecommerce Park, Hanghai Road, Jiubao Town, Jianggan District, Hangzhou City) could be

simply crawled from web pages such as Google Maps[1]. Then leveraging the recently well-studied address entity segmentation methods(Chen et al., 2015), which refers to the process of splitting unstructured standard addresses into address entities,we were able to label the full standard address data with fine-grained entity types with minimal human effort.

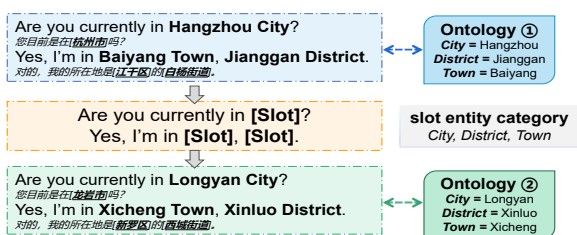

Figure 3: Ontology-based dialogue enrichment.

By using the labeled full standard address, the ontology-based dialogue data enrichment method can expand the limited sampled UR template pairs that include address entities, extracting from current small-size labeled dialogue datasets. Take Figure 3 as an example to explain our enrichment method: "Hangzhou city", "Jianggan District" and "Baiyang Town" in the origin UR pair are replaced by new values in ontology ②. About 578 groups of UR pair templates we used to create dialogue datasets are extracted from the CUCC call center dataset[2] and the Taobao E-commercial Conversation Corpus [3]. The candidate values for each of the ontologies are derived from Tianchi Competitions' full standard address dataset, which has 8.8K full standard addresses with 16 types. More full standard address data can be crawled from websites and labeled using address segmentation methods with limited efforts following (Chen et al., 2015).

### 4.2 Noise injection

To bridge the gap between actual customer support scenarios and ontology-based method-generated dialogues, we present the noise injection method, which captures the typical speaking habits of people in real-life situations. To be specific, we implement five strategies following the prior works (Tian et al., 2021; Feng et al., 2021; Liu et al., 2020): 1) **Pause**, 2) **Repeat**, 3) **Restart**, 4)**Insert**, and 5)**Repair**. The details could be referenced to the above mentioned work. To generate the in-

[1]https://www.google.com
[2]https://www.datafountain.cn/competitions/536/datasets
[3]https://github.com/cooelf/DeepUtteranceAggregation

| | Original texts | Altered texts |
|---|---|---|
| **Sentence level** | U: Please provide the detailed address, where are you now? | U: Could you kindly furnish the specific address of your current location?
U: Can you please share the precise whereabouts of your present position?
U: May I request the elaborate address of your current whereabouts? |
| | R: I am in Hangzhou, Zhejiang Province. | R:My current location is Hangzhou, located in Zhejiang Province.
R:I find myself in Hangzhou, which is situated in Zhejiang Province.
R:I am located in Hangzhou, specifically in Zhejiang Province. |
| **Dialogue level** | U: Repeat please, I didn't hear you clearly.
R: I am in the dental hospital. | U:Repeat please, I didn't catch what you said clearly.
R:I am currently situated in the dental hospital.
U:Could you please repeat that? Your words weren't clear to me.
R:I am currently located at the dental hospital.
U:I apologize for the inconvenience. Could you repeat your statement?
R:I am currently in the dental hospital. |

Table 1: Generated examples from ChatGPT augmentation method.

correct pronunciation, we employ xpinyin [4] and Pinyin2Hanzi [5] to mutually convert pinyin and Chinese characters.

### 4.3 ChatGPT improves diversity

We employ ChatGPT [6], a powerful data augmentation strategy (Ubani et al., 2023), to increase data diversity by improving current UR templates and transforming created dialogues (after Noise injection) into new forms that differ from the original but retain the same meaning.

We have designed prompts for rephrasing at the sentence level (UR pair templates) and dialogue level (after Noise injection). The specific prompt for sentence-level rephrasing is *"Please rewrite the following sentence I give: {text}, and give me 3 rephrased results"*. The number of UR pair templates is increased from 578 to 778 for Ontology-based dialogue enrichment. The prompt for dialogue-level rephrasing is *"Please rewrite the following dialogue I give: {text}, and give me 2 rephrased answers while remaining the number of sentences"*. The amount of generated dataset exceeded 1k after ChatGPT enhancement. Table 1 shows the samples generated using ChatGPT. We also consider the degree of similarity between the new samples and the ones we already have, as illustrated in Figure 4. We are able to better capture data invariance and increase the sample size by leveraging ChatGPT's outstanding generating capabilities.

## 5 Experiments

### 5.1 Datasets

**FGAER-dia** We create a labeled multi-turn dialogue dataset called FGAER-dia, which is constructed according to the paradigm in Section 4.

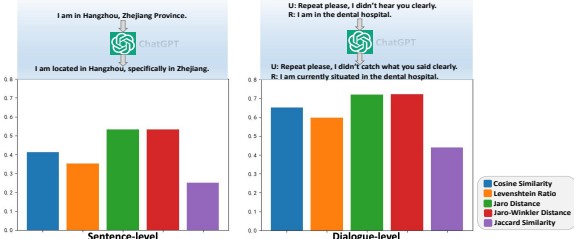

Figure 4: The average Cosine Similarity, Levenshtein Ratio, Jaro Distance, Jaro-Winkler Distance, and Jaccard Similarity for 100 pairs of generated samples and original samples are listed individually at the sentence and dialogue levels.

FGAER-dia is a labeled multi-turn dialogue dataset of 9.8k groups of dialogue contexts with 16 types of fine-grained address entities. The generated dialogue's average length of characters is 167 and the maximum length is 366 in FGAER-dia. The maximum number of turns is 10. Our proposed dataset and the code for our Log-FGAER method are openly available[7].

**CUCC-labeled** The CUCC call-center dialogue dataset provided by DataFountain Competition[2] is transcribed by a real-world downstream speech recognition system, including plenty of transcription errors. We filter out 200 dialogues containing address entities and manually label the filtered data with 16 different types of address entities, to evaluate the Log-FGAER's performance on the multi-turn dialogues from real-life service environment.

Table 5 in Appendix contains more detailed examples of fine-grained address entity types.

### 5.2 Implementation details

In our experiments, ERNIE is performed with BaiDu PaddlePaddle [8], which is a framework of deep learning. For hyperparameter settings, the max sequence length of RoBERTa-wwm-ext,

---

[4]https://pypi.org/project/xpinyin/
[5]https://www.cnpython.com/pypi/pinyin2hanzi
[6]https://openai.com/product/chatgpt

[7]https://github.com/Devil0817/LOG-FGAER
[8]https://www.paddlepaddle.org

| Methods | | FGAER-dia | | | CUCC-labeled | | |
|---|---|---|---|---|---|---|---|
| | | **P** | **R** | **F1** | **P** | **R** | **F1** |
| **BERT-based** | Baseline-1 | 92.32 | 91.09 | 91.55 | 65.72 | 61.72 | 61.75 |
| | Baseline-2 | 92.26 | 91.62 | 91.83 | 72.43 | 67.74 | 68.53 |
| **RoBERTa-based** | Baseline-3 | 90.85 | 90.88 | 90.86 | 64.66 | 64.64 | 63.35 |
| | Baseline-4 | 92.79 | 91.68 | 92.21 | 69.59 | 65.77 | 66.86 |
| | Baseline-5 | 93.60 | 93.23 | 93.41 | 69.77 | 73.37 | 70.54 |
| **ERNIE-based** | Baseline-6 | 92.10 | 92.46 | 92.28 | 83.39 | 83.99 | 83.69 |
| | Baseline-7 | 93.28 | 92.78 | 93.03 | **87.47** | 82.91 | 85.13 |
| | Baseline-8 | 93.15 | 91.68 | 92.41 | 82.10 | 84.17 | 83.12 |
| | **Log-FGAER** | **94.15** | **93.63** | **93.88** | 86.20 | **87.12** | **86.65** |

Table 2: Experiment results on FGAER-dia and CUCC-labeled datasets, and the best performance is highlighted in boldface.

BERT, and ERNIE is 256. The hidden state size of BiLSTM is 128 and the dropout rate is set as 0.1 for BiLSTM output. We use the AdamW optimizer in a mini-batch size of 64 with learning rate $\gamma = 5 \times 10^{-5}$ for each experiment. The weight decay of ERNIE is 0.01. The best $\lambda$ is 0.15. The split of 80% for train, 10% for validation and 10% for test is used. The metrics used for evaluation are precision(P), recall(R) and Micro-F1 metrics. The models with the best validation loss that are selected from all epochs, are saved and utilized for testing. For hardware, we use a four-core CPU and an NVIDIA Tesla V100 GPU.

### 5.3 Baseline models.

We compare the performance of Log-FGAER with several strong baseline methods listed as below.

- **Baseline-1** uses the structure of Bert[9]-CRF.

- **Baseline-2** employs BERT-BiLSTM-CRF as the structure.

- **Baseline-3** adopts Chinese RoBERTa-wwm-ext[10]-CRF as the architecture.

- **Baseline-4** adds a BiLSTM layer between RoBERTa-wwm-ext and CRF layers.

- **Baseline-5** has the same structure as Baseline-4, using a distillation approach to transfer knowledge from labeled full standard addresses to dialogues (Wang et al., 2022).

- **Baseline-6** only finetunes the ERNIE encoder.

- **Baseline-7** is based on the ERNIE-CRF structure.

- **Baseline-8** applies ERNIE-BiLSTM-CRF to deal with FGAER task.

### 5.4 Results and Analysis

The following insights can be drawn from the results as listed in Table 2:

**1)** Compared baselines using the same model structures but different PLM encoders(e.g., Baseline-2, Baseline-4 and Baseline-8 all add BiLSTM-CRF layers on encoders), ERNIE significantly outperforms on FGAER task.

**2)** We compare the state-of-the-art (SOTA) structures (Baseline 1-8) for the FGAER task. On the FGAER-dia dataset, Baseline-7 achieves better F1-score performance. As a result, we combine the proposed Logic-guided method with Baseline-7 to obtain the Log-FGAER.

**3)** With the Logic-guided method enhanced, Log-FGAER outperforms other baselines on both FGAER-dia and CUCC-labeled datasets.

To further display the progress of Log-FGAER, we exclusively list all entity types with an F1-score below 80%. As shown in Table 3 and Figure 5, Log-FGAER can efficiently enhance the F1-scores of almost all entity types that are prone to prediction errors. In particular, the Log-FGAER consistently outperforms Baseline-7 (same structure without PSL) by 3.28 percent over F1-score on entity type "subpoi". These findings demonstrate that the logic inference introduced in our method can capture and utilize geographical hierarchy information contained in the context.

---

[9]https://huggingface.co/bert-base-chinese
[10]https://github.com/ymcui/Chinese-BERT-wwm

| Methods | DISTRICT | COMMUNITY | POI | SUBPOI | INTERSECTION | overall |
|---|---|---|---|---|---|---|
| Baseline-6 | 70.45 | 62.36 | 65.86 | 65.79 | 70.10 | 92.28 |
| Baseline-7 | 70.88 | 62.91 | **66.43** | 67.18 | 70.48 | 93.03 |
| Baseline-8 | 70.13 | 62.30 | 66.13 | 58.07 | 70.50 | 92.41 |
| **Log-FGAER** | **70.98** | **63.12** | 66.28 | **70.46** | **71.38** | **93.88** |

Table 3: F1-scores of Log-FGAER and ERNIE-based baselines on FGAER-dia.

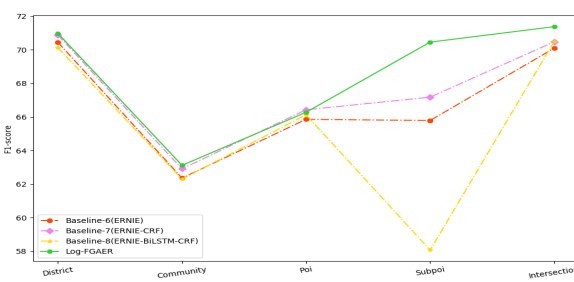

Figure 5: F1-scores of Log-FGAER and ERNIE-based baselines on FGAER-dia.

## 5.5 Ablation Study

Additional experiments were conducted to demonstrate the efficacy of representing contextual information as logic rules that better illustrate the contextual information than comparable methods.

**1)** To begin, we demonstrate how our proposed method can be utilized as an add-on module to improve existing sequence labeling models. We obtain (PSL)ERNIE and (PSL)ERNIE-BiLSTM-CRF after merging the proposed Probabilistic Soft Logic-guided methodology with Baseline-6 (ERNIE) and Baseline-8 (ERNIE-BiLSTM-CRF). As shown in Figure 6, both ERNIE and ERNIE-BiLSTM-CRF improve over various entity types with PSL enhancement. The PSL strategy has proven to be an independent and simple-to-implement module that can be applied to traditional sequence labeling models to make significant improvements.

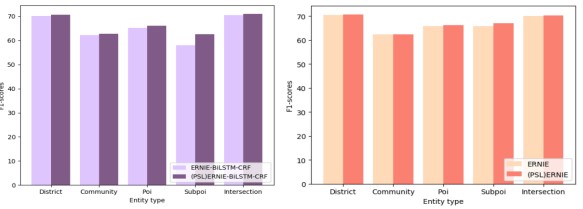

Figure 6: F1 scores of (PSL)ERNIE and (PSL)ERNIE-BiLSTM-CRF models against the same structures without PSL regularization on FGAER-dia.

**2)**Second, we attempt to demonstrate that the PSL strategy expresses contextual information

better than comparable approaches in multi-turn dialogues. We compare to CMV(Luoma and Pyysalo, 2020) and Multi-head Attention methods. **CMV**(Luoma and Pyysalo, 2020) explores cross-sentence contexts for NER with BERT. **Multi-head Attention** employs the ERNIE-CRF structure to produce novel representations with context information. This is accomplished by adding address entities from the preceding and subsequent sequences to the current sequence. Take Figure 2(a) as an example: "Hangzhou City" is added at the front, while "Jiubao Town" is inserted at the end of $S_i$, forming a new representation as the input. In this way, the contextual information could be learned by Transformer multi-head attention mechanism. The comparisons are also evaluated on the FGAER-dia.

Log-FGAER outperforms the other two approaches, as shown in Table 4, demonstrating the efficiency of our logic-guided method in exploiting contextual address hierarchy information.

| Methods | P | R | F1 |
|---|---|---|---|
| **Multi-head attention** | 91.87 | 92.02 | 91.94 |
| **CMV** | 93.09 | 93.17 | 93.12 |
| **Log-FGAER** | **94.15** | **93.63** | **93.88** |

Table 4: Results of models using different methods to obtain contextual information on FGAER-dia.

## 6 Conclusion

In this paper, we propose an effective method named Log-FGAER to extract address entities from spoken dialogue contexts using the PSL rule. Leveraging the spatially constrained relationship of address entities in the dialogues, we improve entity recognition inference by presenting the interdependent hierarchy relationship as the logic rule to regulate the loss objective. We also propose a data augmentation paradigm to construct the labeled multi-turn dialogue dataset for training and evaluation of our proposed method. Experiments on the multi-turn spoken dialogue dataset show the effectiveness of our method.

## Limitations

While our model achieves satisfactory results, it still has several limitations. This study has some potential limitations based on the datasets. Due to the lack of large-scale bench datasets, we created the FGAER-dia and CUCC-labeled datasets, which were both originally constructed in Chinese. In addition, both of these datasets contain a small amount of data. This can lead to an increasing risk of biased or incomplete conclusions and generalizations when applied to real-world scenarios. These challenges and constraints will be addressed in future work.

## Ethics Statement

This work adheres to ethical guidelines and responsible practices in ACL policies. The authors have no external conflicts of interest to declare and have not been required to seek any ethics clearances in order to undertake this work. The datasets used in this work are from previously published works or competitions and, in our view, do not have any attached privacy or ethical issues. We provided data augmentation approaches to improve the evaluation of our proposal. The augmentation-generated conversations aim to provide an optional way to inspire research rather than violate ethical guidelines and should be used in a strictly controlled manner. Transparency and responsible reporting are priorities, and we welcome community engagement to enhance ethical considerations in our work.

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

# A  Appendix

| Label | Interpretation | Example |
|---|---|---|
| PROVINCE | name of a province | 浙江
Zhejiang Province |
| CITY | name of a city | 杭州市
Hangzhou City |
| DISTRICT | name of a district in a city | 西湖区
Xihu District |
| TOWN | name of a town or a boulevard | 九堡镇
Jiubao Town |
| ROAD | name of a road | 航海路
Hanghai Road |
| INTERSECTION | road junction | 交叉口
Intersection |
| ROADNO | road number | 5号
#5 |
| DEVZONE | a economic development zone | 开发区
Development Zone |
| COMMUNITY | name of a community or a village | 宁安社区
Ningan Community |
| POI | name of the point of interest | 衢州人民医院
Quzhou Hospital |
| SUBPOI | name of the second point of interest | 放射科
Radiology Department |
| ASSIST | a phrase indicating relative position | 对面
Opposite |
| DISTANCE | amount of space between two points | 100米
100m |
| HOUSENO | house number | 3幢
Block #3 |
| CELLNO | cell number | 2单元
Unit #2 |
| FLOORNO | floor number | 6层
Level 60 |

Table 5: Interpretations and examples for each entity type used in our datasets, according to the address hierarchical order from top to bottom.