# OpenReview forum: "Log-FGAER: Logic-Guided Fine-Grained Address Entity Recognition from Multi-Turn Spoken Dialogue"
_EMNLP/2023/Conference — EMNLP 2023 Main_

### Official Review · Reviewer_QkBv · 2023-07-25

**Soundness:** 3

**Excitement:**

3: Ambivalent: It has merits (e.g., it reports state-of-the-art results, the idea is nice), but there are key weaknesses (e.g., it describes incremental work), and it can significantly benefit from another round of revision. However, I won't object to accepting it if my co-reviewers champion it.

**Paper Topic And Main Contributions:**

This paper discusses a probabilistic soft logic approach for modeling the hierarchy of parts of an address (e.g., district -> city) during a multi-turn dialogue, which is combined with a pre-trained language model (i.e., ERNIE-CRF) for BIOE labeling of addresses. The contributions include a novel approach for combining a standard cross-entropy loss for a LM with a probabilistic logic model’s loss, as well as several techniques for augmenting small dialogue datasets (i.e., replacing slots in existing templates with different slot values, injecting noise, and generating new templates with ChatGPT).

**Questions For The Authors:**

Question A: Based on the logic rules in Section 3, it seems that a big assumption in this work is that earlier turns in the dialogue have lower-level locations such as a city, while later turns have higher-level locations such as a district. Is this assumption always true? What other sequence-labeling tasks have a similarly hierarchical nature besides addresses?
Question B: What is the computational cost of these experiments (i.e., training and inference)?
Question C: Might you compare BERT-style models to LLM (fine-tuned or few-shot) baselines?
Question D: Is the improvement (3.28% F1) from the probabilistic logic loss significant?
Question E: What is your ethics statement?


**Reasons To Accept:**

The contributions include a novel approach for combining a standard cross-entropy loss for a LM with a probabilistic logic model’s loss, as well as several techniques for augmenting small dialogue datasets (i.e., replacing slots in existing templates with different slot values, injecting noise, and generating new templates with ChatGPT). The paper is well-written, and the experiments comparing to other contextual embedding baselines are thorough and convincing. The code and dataset are open-source.

**Reasons To Reject:**

This is a narrow use case for the technique, specific to detecting addresses with the 16 labels shown in Table 5, for e-commerce assistants delivering purchased items.

**Reproducibility:**

4: Could mostly reproduce the results, but there may be some variation because of sample variance or minor variations in their interpretation of the protocol or method.

**Reviewer Confidence:**

4: Quite sure. I tried to check the important points carefully. It's unlikely, though conceivable, that I missed something that should affect my ratings.

**Typos Grammar Style And Presentation Improvements:**

•	Spell out POI the first time it’s used
•	“spread across to” -> “spread across”
•	“entity containing in” -> “entity contained in”
•	“logic rule that explain” -> “logic rule that explains”
•	Some spaces are missed between text and opening/closing parentheses (section 4)
•	Summarize the noise injection prior work in the appendix
•	Add examples of where probabilistic logic fixes mistakes, with analysis, to the appendix

---

> ### Author Rebuttal · Authors · 2023-08-28
>
> We appreciate the perceptive suggestions provided by reviewers, which will substantially enhance the quality of the paper.
>
> __Question A__: Based on the logic rules in Section 3, it seems that a big assumption in this work is that earlier turns in the dialogue have lower-level locations such as a city, while later turns have higher-level locations such as a district. Is this assumption always true? What other sequence-labeling tasks have a similarly hierarchical nature besides addresses?
>
> __Answer__：
> Thank you for providing us the opportunity to clarify this concern. For the question “is this assumption always true”? The scenarios we considered primarily involved an address-acquiring conversation, such as e-commence shipping, an emergency call for help, registering after-sales services, and so on, in which a full address is frequently formed through a conversation process. According to the statistics we have from real-world dialogues in our system, approximately __82.14%__ of the dialogues follow the top-down geographical hierarchy guideline.
>
> Other tasks, in addition to address entity recognition, use the hierarchical structure rules available in the data to solve their problems. For example, [1] explores the subsumption hierarchy among the relational phrases and constructs a hypernymy graph of phrases using PSL rules. [2] illustrates the hierarchical structure of mobility-related entities, attributes, attribute values, and relations in clinical notes.
>
> _[1]Grycner A, Weikum G, Pujara J, et al. Relly: Inferring hypernym relationships between relational phrases[C]//Proceedings of the 2015 Conference on Empirical Methods in Natural Language Processing. 2015: 971-981._
>
> _[2]Thieu T, Maldonado J C, Ho P S, et al. A comprehensive study of mobility functioning information in clinical notes: entity hierarchy, corpus annotation, and sequence labeling[J]. International journal of medical informatics, 2021, 147: 104351._
>
> __Question B__: What is the computational cost of these experiments (i.e., training and inference)?
>
> __Answer__:
> During the ERNIE fine-tuning stage:
> For an input dialogue sequence of length $N$, the computational complexity is: $O(DLN^2+4SJ)=O(DLN^2+4N)$,
> where D represents the dimension size of the hidden layer and L is the number of layers. The turn of this dialogue is $S$, and the average length of each turn is $J$. We iterate through each turn in the dialogue, searching its two preceding turns and two subsequent turns seperately for address entities ($O(4SJ)$)
>
> During the CRF training stage:
> The complexity of training CRF is $O(NK^2)$,
> where K represents the number of label types.
>
> During the inference stage:
> The complexity is $O(DLN^2+NK^2)$
>
> __Question C__: Might you compare BERT-style models to LLM (fine-tuned or few-shot) baselines?
>
> __Answer__: BERT-style models, if we understand the question right, refer to BERT or Roberta. As illustrated in baselines 1-4, we fine-tuned RoBERTa and BERT using neural network layers (BiLSTM and CRF).
>
> __Question D__: Is the improvement (3.28% F1) from the probabilistic logic loss significant?
>
> __Answer__: We also evaluate the Log-FGAER performance over a small number of the the real-life call-center dialogue data. Our method’s accuracy increased by roughly ___5%___ when compared to an approach with the same structure but no logical rules. We conduct case studies to see how Log-FGAER works: When checking the labeling results, Log-FGAER corrects the entity type of "clinical pharmacy department" in a sentence to "SUBPOI" instead of the previously incorrect type of "COMMUNITY," a mistake that is frequently made. This alteration is consistent with the logical rule we established, which states that the entity type of "clinical pharmacy department" should be located between 'POI' and 'HOUSENO.' This real-world example demonstrates the efficacy of our method.
>
> __Question E__: What is your ethics statement?
>
> __Answer__: We will include the following ethics statement at the end of the paper:
> This work adheres to ethical guidelines and responsible practices in ACL policies. The authors have no external conflicts of interest to declare and have not been required to seek any ethics clearances in order to undertake this work. The datasets used in this work are from previously published works or competitions and, in our view, do not have any attached privacy or ethical issues. We provided data augmentation approaches to improve the evaluation of our proposal. The augmentation-generated conversations aim to provide an optional way to inspire research rather than violate ethical guidelines and should be used in a strictly controlled manner. Transparency and responsible reporting are priorities, and we welcome community engagement to enhance ethical considerations in our work.

---

### Official Review · Reviewer_ke6G · 2023-08-04

**Soundness:** 4

**Excitement:**

4: Strong: This paper deepens the understanding of some phenomenon or lowers the barriers to an existing research direction.

**Paper Topic And Main Contributions:**

The paper tackles the problem of fine-grained address entity recognition (FGAER) from multi-turn dialogs, which is challenging due to the presence of noise in spoken dialogs and addresses often spanning multiple turns. The paper proposes using probabilistic soft logic (PSL) to modify the loss function to encode the spatial dependence of addresses in a dialog, which is shown to outperform competitive baselines in the experiments. The proposed enhancement can be combined with any existing FGAER models. To address the lack of labeled dialogs with address entities, the paper proposes an ontology-based data augmentation with ChatGPT to generate additional data.

**Questions For The Authors:**

A. Is separating fine-tuning PLM encoder with PSL regularization from training CRF a deliberate choice and your recommendation vs training PLM-CRF with PSL regularization together? I.e., do you recommend adding PSL regularization only to fine-tuning PLM instead of training the full model end-to-end?

**Reasons To Accept:**

- The proposed logic-guided enhancement for FGAER is well-motivated, novel, and general. Log-FGAER can be combined with any existing FGAER model without modifying model architecture, making it easy to adopt in practice. The proposed method is also not limited to encoding spatial dependence of addresses; other domain knowledge could be encoded with PSL in a similar way to improve NER performance, making the method an effective way to combine domain knowledge with data-driven approaches.
- Even though data augmentation with LLMs, especially for paraphrasing, is not new, the proposed end-to-end data augmentation approach leveraging crawled addresses, entity segmentation method, and ChatGPT is a practically useful addition.

**Reasons To Reject:**

- It would be good to see some analyses justifying the hierarchy relationship among addresses in a dialog. The main relationship encoded with PSL assumes that the scope of location decreases as dialog progresses, but users can easily break such hierarchy, and the conventional order of providing addresses may change depending on the language (e.g., Chinese vs English). It’s not a huge concern, however, as the proposed method is general enough to encode other relationships among entities.
- In the experiments, the proposed Log-FGAER adds the proposed logic-guided enhancement to Baseline-7. Since Baseline-7 is already mostly outperforming the rest of the baselines, comparing Log-FGAER with the rest of the baselines is not very informative. A more informative experiment would be to compare each baseline’s performance with and without the proposed enhancement (basically expand the first ablation study to other baselines in the main experiments). Such experiment can better support the claim that the proposed enhancement can be combined with any existing FGAER method.

**Reproducibility:**

4: Could mostly reproduce the results, but there may be some variation because of sample variance or minor variations in their interpretation of the protocol or method.

**Reviewer Confidence:**

4: Quite sure. I tried to check the important points carefully. It's unlikely, though conceivable, that I missed something that should affect my ratings.

**Typos Grammar Style And Presentation Improvements:**

- Line 54 - Not clear why the referenced methods are not considered due to their labels, since I assume you can still use the same model architectures and train on your own labeled data.
- Line 148-150 – Similarly, not clear why those methods are not applicable just because of different entity types.
- Line 181-182 – a nitpick on the notations: typically upper case letters are random variables and lower case letters are realized instances of these random variables.
- Line 396 – Referring the implementation details to survey papers is not very helpful; it’d be better to just have the implementation details in the appendix.
- Line 418 – nitpick: “retaining” would be a better choice of word than “remaining”
- Figure 5 – nitpick: not sure if it makes sense to draw lines in this figure, since there’s no particular order with the entities on x-axis; better to just keep the dots or draw barplots.

---

> ### Author Rebuttal · Authors · 2023-08-25
>
> We appreciate the reviewer's time, which will substantially improve this paper.
>
> __Question 1__: Is separating fine-tuning PLM encoder with PSL regularization from training CRF a deliberate choice and your recommendation vs training PLM-CRF with PSL regularization together? I.e., do you recommend adding PSL regularization only to fine-tuning PLM instead of training the full model end-to-end?
>
> __Answer__: We appreciate your professional inquiry. As a matter of fact, we have conducted the experiment of training PLM-CRF with PSL regularization together, and the result is not as good as first fine-tuning the PLM encoder with PSL regularization before training the CRF layer. This result will be included in the revision. Our findings show that fine-tuning the PLM encoder with PSL regularization can capture address information, which might aid the subsequent CRF layer in geographical tag decoding.
>
> __Question 2__：It would be good to see some analyses justifying the hierarchy relationship among addresses in a dialog. The main relationship encoded with PSL assumes that the scope of location decreases as dialog progresses, but users can easily break such hierarchy, and the conventional order of providing addresses may change depending on the language (e.g., Chinese vs English). It’s not a huge concern, however, as the proposed method is general enough to encode other relationships among entities.
>
> __Answer__: Thank you for providing us the opportunity to clarify this concern. The scenarios we considered primarily involved an address-acquiring conversation, such as e-commence shipping, an emergency call for help, registering after-sales services, and so on, in which a full address is frequently formed through a conversation process. According to the statistics we have from real-world dialogues in our system, approximately ___82.14%___ of the dialogues follow the top-down geographical hierarchy guideline.
> In addition, this method is not limited to a particular address naming system (such as Chinese) but could also be applied to other addressing systems that follow certain geographic location rules, according to references [1–3].
>
> _[1]Manoruang D, Asavasuthirakul D. Quality analysis of online geocoding services for Thai text addresses[J]. Engineering and Applied Science Research, 2019, 46(2): 86-97._
>
> _[2]Wang X, Zhang Y, Chen M, et al. An evidence-based approach for toponym disambiguation[C]//2010 18th International Conference on Geoinformatics. IEEE, 2010: 1-7._
>
> _[3]Naaman M, Song Y J, Paepcke A, et al. Automatic organization for digital photographs with geographic coordinates[C]//Proceedings of the 4th ACM/IEEE-CS joint conference on Digital libraries. 2004: 53-62._
>
> __Quesion 3__: In the experiments, the proposed Log-FGAER adds the proposed logic-guided enhancement to Baseline-7. Since Baseline-7 is already mostly outperforming the rest of the baselines, comparing Log-FGAER with the rest of the baselines is not very informative. A more informative experiment would be to compare each baseline’s performance with and without the proposed enhancement (basically expand the first ablation study to other baselines in the main experiments). Such experiment can better support the claim that the proposed enhancement can be combined with any existing FGAER method.
>
> __Answer__: Thank you for your useful suggestions. We will incorporate the ablation study into the revision.

---

### Official Review · Reviewer_NiYR · 2023-08-05

**Soundness:** 3

**Excitement:**

3: Ambivalent: It has merits (e.g., it reports state-of-the-art results, the idea is nice), but there are key weaknesses (e.g., it describes incremental work), and it can significantly benefit from another round of revision. However, I won't object to accepting it if my co-reviewers champion it.

**Paper Topic And Main Contributions:**

This paper addresses the problem of fine-grained address entity extraction from multi-turn dialogs in Chinese. The proposed approach improves the loss function of an ERNIE-based model by relying on probabilistic soft logic to consider information regarding the administrative division hierarchy and the way addresses are typically provided in a dialog. Additionally, several augmentation methods are used to obtain more data for training the model: the address entities in existing dialogs are replaced by other entities of the same type, noise is injected in the dialogs to simulate speaking habits in real-world conversations, and ChatGPT is used generate variations of the dialogs. The proposed approach achieves 93.88 F1 score on a dataset generated using this augmentation approach and 86.65 on a real-world dataset, surpassing several competitive baselines.

**Reasons To Accept:**

Automatically extracting addresses from a dialog is an important step for automating processes such as online ordering and emergency dispatching. Thus, this paper addresses a relevant problem that is not widely regarded in the literature. The proposed approach is compared to several strong baselines and the discussion of the results is thorough, including analysis per entity type.

**Reasons To Reject:**

As stated in Definition 6, the proposed approach assumes that addresses become more fine-grained along the dialog, that is, an address entity type that is higher in the administrative division hierarchy does not appear in a dialog turn subsequent to one that includes an address entity type that is deeper in the hierarchy (e.g the province is not provided after the city). I'm not fluent in Chinese, however, I believe that this may not be always the case, independently of the language. For instance, a broader entity type can be used for disambiguation purposes. Thus, this may be a flaw in the approach that impacts its performance in real conversations.

**Reproducibility:**

3: Could reproduce the results with some difficulty. The settings of parameters are underspecified or subjectively determined; the training/evaluation data are not widely available.

**Reviewer Confidence:**

3: Pretty sure, but there's a chance I missed something. Although I have a good feel for this area in general, I did not carefully check the paper's details, e.g., the math, experimental design, or novelty.

**Typos Grammar Style And Presentation Improvements:**

The citation format when the reference is used as the subject is not correct. For instance, in line 127, instead of "(Mai et al., 2018) improves", it should read Mai et al. (2018) improves

(039) "further delivery" makes no sense here

(Figure 1) "mui" -> multi

(131) "with us" -> to ours

(396) "could be referenced to" -> are described in

(411-419) The "I give" and "remaining" parts of the prompts sound weird

(419) dataset -> data

---

> ### Author Rebuttal · Authors · 2023-08-25
>
> We appreciate the reviewer's insightful suggestions, which will significantly improve the paper.
>
> __Question__: As stated in Definition 6, the proposed approach assumes that addresses become more fine-grained along the dialog, that is, an address entity type that is higher in the administrative division hierarchy does not appear in a dialog turn subsequent to one that includes an address entity type that is deeper in the hierarchy (e.g the province is not provided after the city). I'm not fluent in Chinese, however, I believe that this may not be always the case, independently of the language. For instance, a broader entity type can be used for disambiguation purposes. Thus, this may be a flaw in the approach that impacts its performance in real conversations.
>
> __Answer__: Thanks for giving us the chance to clarify this concern. There are different kinds of conversations, even goal-driven dialogue. Yes, it may not always be the case that the fine-grained address entity mentioned follows the hierarchy during the dialog turn. For example, an airport booking conversation may only mention the city. As a result, as explained in the Introduction section, the scenarios we considered primarily involved an address-acquiring conversation, such as e-commence shipping, an emergency call for help, registering after-sales services, and so on, in which a full address is frequently formed through a conversation process. These scenarios are quite diverse, and it is extremely beneficial to assist in automatically collecting fine-grained address entities, such as in an emergency.  According to the statistics we have from real-world dialogues in our system, approximately 82.14% of the dialogues follow the top-down geographical hierarchy guideline.
>
> In addition, our method is not limited to a particular address naming system but could also be applied to other addressing systems that follow certain geographic location rules, according to references [1–3].
>
> _[1]Manoruang D, Asavasuthirakul D. Quality analysis of online geocoding services for Thai text addresses[J]. Engineering and Applied Science Research, 2019, 46(2): 86-97._
>
> _[2]Wang X, Zhang Y, Chen M, et al. An evidence-based approach for toponym disambiguation[C]//2010 18th International Conference on Geoinformatics. IEEE, 2010: 1-7._
>
> _[3]Naaman M, Song Y J, Paepcke A, et al. Automatic organization for digital photographs with geographic coordinates[C]//Proceedings of the 4th ACM/IEEE-CS joint conference on Digital libraries. 2004: 53-62._

---

### Meta-Review · Area_Chair_DDHY · 2023-09-26

**Recommendation:** 4

**Metareview:**

The paper proposes a method for logic-guided fine-grained address recognition method, where address hierarchy has been utilized as the logic rule and apply this in a probabilistic manner to improve the accuracy. Automatic extraction of addresses from conversation is a nice step. The paper needs revision to address the following issues: ablation and qualitative analysis to demonstrate the justification of hierarchy relationships of addresses; ablation by including logic rules to each of the baselines.

---

### Decision · Program_Chairs · 2023-10-07

**Decision:**

Accept-Main

**Comment:**

The paper proposes a method for logic-guided fine-grained address recognition method, where address hierarchy has been utilized as the logic rule and apply this in a probabilistic manner to improve the accuracy. Automatic extraction of addresses from conversation is a nice step. The paper needs revision to address the following issues: ablation and qualitative analysis to demonstrate the justification of hierarchy relationships of addresses; ablation by including logic rules to each of the baselines.